# Faster and Better: How Anomaly Detection Can Accelerate and Improve Reporting of Head Computed Tomography

**DOI:** 10.3390/diagnostics12020452

**Published:** 2022-02-10

**Authors:** Tom Finck, Julia Moosbauer, Monika Probst, Sarah Schlaeger, Madeleine Schuberth, David Schinz, Mehmet Yiğitsoy, Sebastian Byas, Claus Zimmer, Franz Pfister, Benedikt Wiestler

**Affiliations:** 1Department of Diagnostic and Interventional Neuroradiology, Klinikum Rechts der Isar, Technische Universität München, Ismaninger Straße 22, 81675 Munich, Germany; monika.probst@tum.de (M.P.); sarah.schlaeger@tum.de (S.S.); madeleine.schuberth@tum.de (M.S.); david.schinz@tum.de (D.S.); claus.zimmer@tum.de (C.Z.); b.wiestler@tum.de (B.W.); 2DeepC GmbH, Atelierstraße 29, 81671 Munich, Germany; julia.moosbauer@deepc.ai (J.M.); mehmet@deepc.ai (M.Y.); sebastian.byas@deepc.ai (S.B.); franz.pfister@deepc.ai (F.P.)

**Keywords:** machine learning, neuroradiology, computed tomography, decision support, anomaly detection, classification

## Abstract

Background: Most artificial intelligence (AI) systems are restricted to solving a pre-defined task, thus limiting their generalizability to unselected datasets. Anomaly detection relieves this shortfall by flagging all pathologies as deviations from a learned norm. Here, we investigate whether diagnostic accuracy and reporting times can be improved by an anomaly detection tool for head computed tomography (CT), tailored to provide patient-level triage and voxel-based highlighting of pathologies. Methods: Four neuroradiologists with 1–10 years of experience each investigated a set of 80 routinely acquired head CTs containing 40 normal scans and 40 scans with common pathologies. In a random order, scans were investigated with and without AI-predictions. A 4-week wash-out period between runs was included to prevent a reminiscence effect. Performance metrics for identifying pathologies, reporting times, and subjectively assessed diagnostic confidence were determined for both runs. Results: AI-support significantly increased the share of correctly classified scans (normal/pathological) from 309/320 scans to 317/320 scans (*p* = 0.0045), with a corresponding sensitivity, specificity, negative- and positive- predictive value of 100%, 98.1%, 98.2% and 100%, respectively. Further, reporting was significantly accelerated with AI-support, as evidenced by the 15.7% reduction in reporting times (65.1 ± 8.9 s vs. 54.9 ± 7.1 s; *p* < 0.0001). Diagnostic confidence was similar in both runs. Conclusion: Our study shows that AI-based triage of CTs can improve the diagnostic accuracy and accelerate reporting for experienced and inexperienced radiologists alike. Through ad hoc identification of normal CTs, anomaly detection promises to guide clinicians towards scans requiring urgent attention.

## 1. Introduction

Advances in machine learning (ML) architectures, coupled with the broader availability of digitalized data and improvements of computational hardware, have allowed for artificial intelligence (AI) tools to match and in some cases even surpass human performance [1,2]. Many tasks in radiology, from the acceleration of image acquisition, to the post hoc reduction of compromising artifacts and detection of various pathologies can be improved through innovative AI techniques [3,4,5]. Despite having been validated in their respective test settings, many AI systems are nonetheless still confronted with high skepticism as it oftentimes remains unclear if their performance is confined to the environments in which they have been tested in or if they can be of true, real-world utility. In radiology, AI solutions must furthermore be tailored toward a meaningful use-case where they ideally perform a task that is either time-consuming or error-prone for the radiologist to do. Analogous to the pilot–autopilot liaison, a human–machine interface ideally emerges where laborious tasks such as data screening are performed by the AI while decision-making remains confided to the human expert.

Multislice imaging in particular has recently witnessed the deployment of various decision-support systems, mainly aimed at flagging emergency findings, such as large-vessel occlusion in suspected stroke patients, or screening procedures such as CT-based lung cancer detection [6,7]. While excellent diagnostic performance was reported for these systems, they mainly consider single-class segmentations and thus remain of limited clinical utility in an unselected cohort. Multi-class systems, on the other hand, have been proposed for less data-intensive imaging modalities, such as chest X-ray and hip radiographs, that are generally readily interpretable by the radiologist [8,9].

Previously, we have reported on the utility of an anomaly detection tool aimed at identifying pathology in routinely acquired head CTs and providing subsequent triage based on the lack or presence of a suspected pathology [10,11]. Importantly, this tool has been designed to detect all types of intracranial pathologies and needs only weak supervision during its training phase, circumventing the problem that supervised AI tools require vast amounts of laboriously annotated data and can only detect what they have previously been trained to “see”. To elucidate the potential role of this anomaly-detection system in clinical routine, we here investigate the performance of neuroradiologists at different experience levels asked to screen head CT scans both with and without the AI support. We were particularly interested to compare the completeness of reports as well as the reporting times for both approaches, hypothesizing that an efficient AI tool improves both.

## 2. Materials and Methods

### 2.1. Study Dataset

For analysis, 80 head CT scans (1 scan/patient) acquired in the neuroradiology department of a university hospital in March 2021 were included. The same hardware (Philips Ingenuity 5000, Philips Medical Systems, Best, The Netherlands) was used in all patients with local postprocessing according to a manufacturer-specific iterative model reconstruction (IMR3). Clinical reports, co-signed by at least two neuroradiologists, were used as ground truth to define pathology labels and were additionally validated (using all available information, including follow-up examinations) by the study coordinator (T.F.), not taking part in the study-specific readings. In order to provide a representative case mix, the study dataset consisted of 40 normal scans and 40 scans with various pathological findings. In patients showing pathology, all pathology classes were documented, i.e., if a patient had an intraparenchymal hemorrhage with accompanying intraventricular hemorrhage and midline shift, the intraparenchymal hemorrhage was noted as main finding and intraventricular hemorrhage as well as midline shift were noted as additional findings.

In decreasing order, the most common main findings in pathological scans were: tumor (n = 10), subacute stroke (n = 7), intraparenchymal hemorrhage (n = 6), subarachnoid hemorrhage (n = 5), late stroke (n = 4), acute subdural hematoma (n = 3), cavernoma (n = 2), acute stroke (n = 2) and atrophy (n = 1).

As some scans had multiclass pathology labels, the mean number of findings per pathological scan was 2.2.

### 2.2. Anomaly Detection Tool

The herein investigated anomaly detection tool was trained using a weakly supervised machine learning strategy that only requires a globally annotated dataset, as more elaborately explained in [10]. In short, the network was trained with a dataset of 191 normal head CT scans. In the training stage, all images were registered to an internal template to establish voxel-wise correspondence and define an internal reference atlas. Per-voxel Gaussian density distributions were fitted across the training dataset. Next, an internal reference atlas of a normal brain was defined by calculating the upper and lower bounds for the density distributions mentioned above. ANTs (Advanced Normalization Tools) framework was utilized for the multi-stage registration of images from the study set to the template image of the internal atlas [12].

Outlier voxels were identified by comparing regions in the study set against the voxel-wise upper and lower bounds of the confidence interval from the training step. The ratio of outlier voxels to the entire brain volume was used to determine the patient level anomaly score, ranging from 0 to 1. Based on this anomaly score, patients were categorized into three classes (“normal”, “inconclusive”, “pathological”) via thresholding. These thresholds were scanner-specific and have previously been calibrated on an independent validation set (not included in this study) of 61 CT scans (globally labelled as “normal” or “pathological”) from the local CT scanner to minimize the false positive rate under the constraint of a false omission rate of 0, as explained in [10]. If the anomaly score was higher than the pre-determined upper threshold (s > T-upper), the scan was labelled as “pathological” and anomalous voxels were added to the heatmap. Anomaly scores below the pre-determined lower threshold (s < T-lower), on the other hand, translated into “normal” labels for the scan, while anomaly scores between T-upper and T-lower led to an “inconclusive” label.

Finally, a worklist with all patients of the study set was presented to the radiologist. AI-support provided patient-level predictions into (a) normal (marked in green), (b) inconclusive (marked in white), (c) pathological (marked in red) as well as anomaly maps with the outlier voxels to provide image-based guidance towards an underlying pathology (Figure 1).

### 2.3. Study Setting

Between June 2021 and September 2021, four board-certified neuroradiologists with varying levels of experience (R1: 10 years of experience; R2: 7 years of experience; R3: 2 years of experience; R4: 1 year of experience) rated the 80 CT scans from the study set. Experienced (R1 and R2) and inexperienced (R3 and R4) readers were pooled for analysis. The four neuroradiologists independently rated the scans on the same workstation in two runs, with and without the support of the AI tool. The reporting interface for both runs is shown in Figure 1. Aside from the color-based patient-level predictions into “normal”/“inconclusive”/“pathological” as well as the voxel-based anomaly maps that were overlaid to the DICOM images, all parameters in the software interface were chosen identical irrespective if AI-support was provided or not. To best standardize the reporting process, the readers were asked to attribute labels from a predefined list containing all pathologies contained in the study set or a “normal” label. Reporting times were documented in-app and defined as the time (in seconds) between the opening of a CT study and the submission of the report. Readers were made aware of this time-tracking method in order to prevent interruptions during the reporting task.

To prevent a reminiscence effect, a wash-out period of 4 weeks between both runs was included. For randomization purposes, R2 and R3 first reported with AI-support, while R1 and R4 first reported without AI-Support. The study workflow is shown in Figure 2.

### 2.4. Statistical Analysis

Sensitivities, specificities, positive predictive values (PPV) and negative predictive values (NPV) were calculated based on the ratings given by the readers with/without AI-support and the underlying ground truth labels. Reporting times for the overall study dataset as well as discriminated for normal/pathological scans were calculated as a function of AI-support.

Subjective diagnostic confidence to provide correct patient-level classification into “normal” or “pathological” was assessed through a 5-point Likert scale (1 = very low; 2 = low; 3 = neutral; 4 = high; 5 = very high). Finally, diagnostic completeness, i.e., the ability to identify all pathology classes mentioned in the reference radiological report was compared between runs.

Continuous variables were reported as mean ± SD, while discrete variables were given as median with their respective 25% and 75% percentiles. Differences in diagnostic accuracy, and diagnostic confidence were compared with a Wilcoxon signed rank test. Differences in reporting times were compared with a paired Student’s *t* test. Statistical analysis was performed using Graphpad Prism Version 8.4.3. for Mac OS (GraphPad Software, San Diego, CA, USA). *p* values below 0.05 were considered statistically significant.

## 3. Results

### 3.1. System Performance

The AI tool provided definite ratings into “normal” or “pathological” in 60/80 scans, leading to a test yield of 75%. Specifically, definite ratings were given in 20/40 (50%) normal scans and 40/40 (100%) scans showing pathology. Accordingly, all inconclusively labelled scans were normal on ground truth. System predictions in the 60 conclusively labelled scans all corresponded to ground truth. In detail, there was no false-positive label (normal scan erroneously labelled as “pathological”) and no false-negative label (pathological scan erroneously labelled as “normal”).

### 3.2. Diagnostic Accuracy

For the 320 evaluated scans (80 scans/reader), misclassification into “normal” or “pathological” occurred in 11/320 cases (10 false-positive classifications and 1 false-negative classification) without AI support and 3/320 cases (all false-positive classifications) with AI support (*p* for difference: 0.0045). In detail, experienced radiologists reduced their error counts from 4/160 scans to 0/160 scans if supporting predictions were given, while inexperienced readers reduced their error counts from 7/160 to 3/160 with AI support. This translated to a sensitivity, specificity, positive predictive value and negative predictive value of 99.4%, 93.8%, 94.1% and 99.3% if no AI-support was given and 100%, 98.1%, 98.2% and 100% if support by the anomaly detection tool was available. Figure 3 illustrates the classification completeness for both runs.

Of all 14 scans (11 scans and 3 scans, respectively, in both runs) that were erroneously labelled, 13 were normal on ground truth. Only one pathological scan showing acute ischemia as main finding was misclassified as “normal” by an inexperienced reader in the run without AI-support.

Beyond the patient-level labels, diagnostic completeness regarding all relevant underlying pathologies was investigated. In total, 89 pathologies were present in the 40 CT scans showing pathology (mean of 2.2 pathology classes per scan), translating to 356 (4 × 89) pathology labels that could potentially be depicted by the four neuroradiologists. A total of 30/356 pathological findings were missed without AI support, while 17/356 were missed with AI support (*p* = 0.0005). In decreasing order, missed pathologies were: subarachnoid hemorrhage (n = 8), late stroke (n = 7), skull fracture (n = 4), tumor (n = 3), acute stroke (n = 3), acute subdural hematoma (n = 2), intraparenchymal bleed (n = 2) and atrophy (n = 1).

Both, inexperienced (20/356 vs. 11/356, *p* = 0.0015) and experienced (10/356 vs. 6/356, *p* = 0.045) readers experienced significant gains in diagnostic completeness when provided with the anomaly maps and patient labels.

Inexperienced radiologists missed relevantly more pathologies both, with (11/356 vs. 6/356, *p* = 0.025) and without (20/356 vs. 10/356, *p* = 0.0009) AI support compared to experienced readers. Table 1 provides information on classification completeness for all runs and experience levels.

### 3.3. Diagnostic Confidence

In both experienced and inexperienced readers, subjectively assessed diagnostic confidence to provide patient-level labels was similar with and without support of the AI tool. In detail, inexperienced readers reported diagnostic confidence levels of 4.30 ± 0.84 vs. 4.35 ± 0.79 (*p* = 0.71), while experienced readers reported similarly high levels of 4.58 ± 0.74 vs. 4.61 ± 0.72 (*p* = 0.60).

### 3.4. Reporting Speed

Support of the anomaly detection tool allowed for accelerated reporting with a mean reduction in reporting times from 65.1 ± 8.9 s to 54.9 ± 7.1 s (*p* = 0.0001). Of note, the time gains were more pronounced in scans that were normal on ground truth (59.6 ± 7.8 s vs. 46.3 ± 5.0 s, *p* = 0.0001) compared to pathological scans (70.3 ± 10.3 s vs. 63.3 ± 8.6 s, *p* = 0.016). Improvements in reporting speed were comparable between experienced and inexperienced radiologists at rates of 17.0% and 13.9%, respectively. Interestingly, the cohort that benefitted most from the availability of patient-level predictions and pixel-wise anomaly maps were inexperienced radiologists confronted with normal CT scans (62.3 ± 8.4 s vs. 46.3 ± 2.7 s, *p* = 0.0001). There was no subgroup of experience-level and ground truth label where reporting times increased in the run with AI support.

Specific analysis of the scans that were inconclusively labelled revealed comparable reporting times in the runs with (55.5 ± 3.2 s) and without (56.3 ± 2.5 s) AI-support (*p* = 0.76). In scans that were conclusively (“normal”/“pathological”) labelled by the algorithm, reporting times were significantly lower with AI-support (68.3 ± 4.3 s vs. 54.6 ± 3.1 s, *p* = 0.01). Detailed metrics on reporting times are provided in Table 2. Figure 1 furthermore illustrates how overall reporting times benefitted from AI-support.

## 4. Discussion

The purpose of this study was to investigate the clinical utility of an anomaly-detection system aimed at flagging pathology in head CT and providing patient-level triage into “normal” or “pathological”. We found that (i) diagnostic accuracy could be improved, (ii) reporting times were faster and (iii) the subjective confidence levels remained high but unchanged if AI support was provided to neuroradiologists at varying levels of experience.

There is general agreement that ML will contribute greatly to future developments in medical research [13,14]. Although algorithms meant to support radiologists in various tasks have been presented in the past, many of them fail to gain traction in daily use [15,16]. Oftentimes, the reason for this is a lacking “know-the-user” approach where the specific use-case of an ML software remains opaque and seamless integration into the clinical workflow is therefore unrealistic. Contrary to many other tasks in medicine where a target metric can only be approximated, radiological reporting can be elegantly subdivided into the key components of (a) detecting or excluding pathology and (b) doing so in a time-efficient manner, making it a very suitable domain to assess the performance of an AI tool trying to improve both steps.

Our results show that using an anomaly-detection tool aimed at detecting pathologies in raw CT data and providing patient-level labels (“normal” or “pathological”) as well as voxel-level segmentations significantly improved the diagnostic accuracy of both, inexperienced and experienced neuroradiologists. Near perfect positive and negative predictive values of 98.2% and 100% thus became achievable. Importantly, no false-positive or false-negative predictions were made by the algorithm, weakening the argument that a human reader might be biased towards error by inadequate performance of AI tools. While these findings will need to be validated in more extensive, multicenter datasets, our analysis confirms that medical imaging might benefit from the wider implementation of anomaly detection as rates of missed findings as high as 30% have consistently been reported [17].

Interestingly, the here-investigated tool was especially helpful for correctly identifying normal-appearing scans, as the false-positive rate was cut by two-thirds once predictions were given. This seems intuitive as it is known that junior radiologists in particular have the tendency to oversensitively read image data, with potential therapeutic consequences or unnecessary follow-up imaging [18]. Anomaly detection could alleviate this weak point as our tool has been trained to internalize common physiological variances and hence does not flag them as pathology. Even if concrete numbers vary, up to 80% of all routinely acquired head CT scans are normal-appearing [19]. In light of this, the reduction of false-positive ratings by 2/3 with AI support is of particular interest and could prospectively improve the diagnostic accuracy in routine imaging. It should be noted that the here-investigated anomaly detection tool provided definite predictions in only 75% of cases. Through innovative approaches, such as reinforcement learning, the diagnostic yield and hence clinical utility of such systems could be further augmented in the future.

The here-presented AI tool does not provide semantic information, i.e., it does not classify pathologies into different entities (as in ischemia, bleeds, tumor, …). This could help explain why the availability of predictions did not augment diagnostic confidence levels, given that the overlooking of relevant findings remains the main cause for misreporting but is logically not suspected by the radiologist. This observation further builds the case for implementing anomaly detection in radiology as lowering the rate of missed findings should be a natural priority for reducing medical errors and costly litigation cases.

The ongoing technologization in medicine explains why imaging continues to become an even more integral part in clinical decision-making, with the number of imaging exams worldwide inching up to 10^9^/year [20,21]. In addition to the heightened complexity of multislice methods, as in CT and MRI, tackling the disparity between the limited number of radiologists and rising workload is becoming a growing concern in patient care and workplace satisfaction. Thus, solidifying the radiologist’s expert role by supplying triaged exams and segmentations of potential findings seems a promising approach. In many areas, from route proposition by plane autopilots to automated wildfire detection on satellite images, ML can reliably compute laborious tasks in the background without undercutting the decision-making role of the human operator [22]. In comparison, it becomes evident that radiology has to catch up if streamlined integration of software solutions into patient care truly is one of the objectives for the intermediate future. Our results confirm that AI support can lead to significant productivity gains, as reporting could be accelerated by up to 25% in our study set. Coupled with the fact that accelerated pathology detection allows the radiologist to focus more on the intellectually stimulating interpretation task, such advances might hold one of the keys to counteract on deteriorating working conditions that threaten the physical and psychological well-being of healthcare workers [23,24]. Beyond the obvious socioeconomic benefits that could arise from a more time-efficient use of highly paid physicians, the ability of radiologists to focus on the scans requiring urgent and extensive attention could be of obvious value for patient care.

## 5. Conclusions

In conclusion this study validates the clinical utility of an anomaly detection tool for head CTs. By providing a combination of patient-level triage and pixel-wise segmentations of underlying pathology, diagnostic accuracy and reporting times can be significantly improved, in experienced and inexperienced radiologists alike.

## Figures and Tables

**Figure 1 diagnostics-12-00452-f001:**
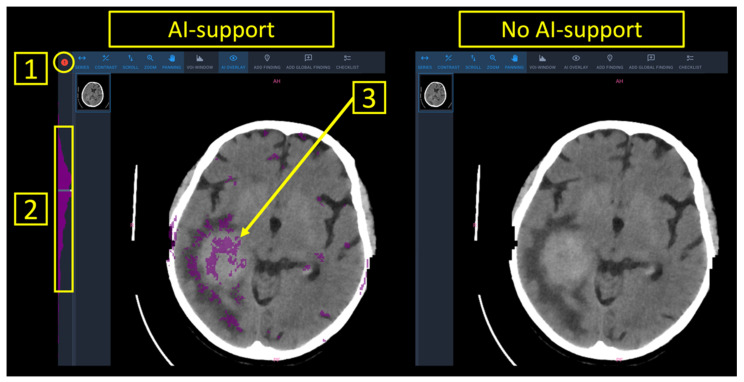
Reporting interface with and without AI support. Note that AI support provided patient-level predictions “normal” (green), “inconclusive” (white) or “pathological” (red), as highlighted by (**1**). Furthermore, pixel-wise segmentations of suspected pathology (**3**), as well as the distribution of anomalous pixels within the stack of CT images (**2**) were available.

**Figure 2 diagnostics-12-00452-f002:**
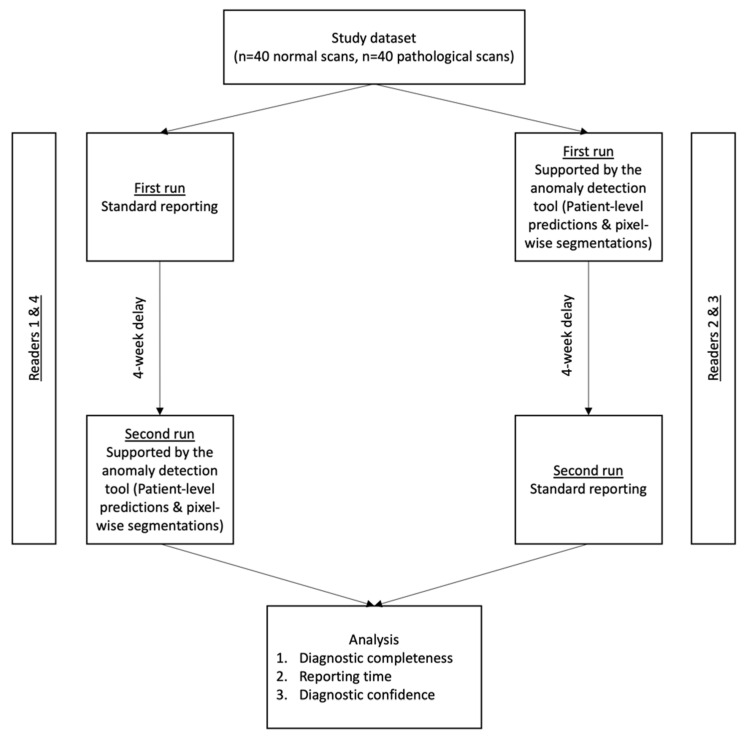
Study workflow. The test set consisted of 80 head CTs (40 normal scans and 40 scans showing common intracranial pathologies, as elaborated in the methods section). All readers completed a run with and without AI-support, in a randomized and alternating order. Analyzed endpoints were (i) the diagnostic accuracy to discriminate between normal and pathology-showing CT, (ii) reporting times and (iii) the subjectively assessed diagnostic confidence in patient-level labels.

**Figure 3 diagnostics-12-00452-f003:**
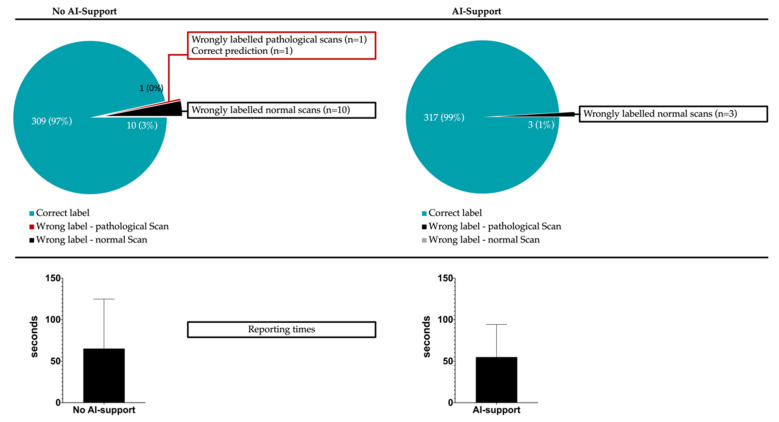
Given is the classification completeness with/without AI-support (upper panel). Patient-level misclassification into “normal” or “pathological” could be significantly reduced from 11/320 cases to 3/320 cases (*p* < 0.0001) once AI support was available. The lower panel gives the mean (± standard deviation) reporting time per scan with (54.9 ± 7.1 s) and without (65.1 ± 8.9 s) AI support (*p* < 0.0001).

**Table 1 diagnostics-12-00452-t001:** Classification completeness for experienced and inexperienced readers with/without AI-support.

	No AI-Support	AI-Support
**Patient-level misclassifications**		
All	11/320	3/320
Experienced	4/160	0/160
Inexperienced	7/160	3/160
**Number of pathologies missed**		
All	30/356	17/356
Experienced	10/356	6/356
Inexperienced	20/356	11/356

**Table 2 diagnostics-12-00452-t002:** Reporting times (mean ± standard deviation) in seconds for experienced/inexperienced readers, as well as subgroups according to experience levels and ground truths. GT: ground truth, RT: reporting time.

Subgroups	No AI Support (s)	AI Support (s)	D RT	*p*
All	65.1 ± 8.9	54.9 ± 7.1	15.7%	0.0001
All—GT Normal	59.6 ± 7.8	46.3 ± 5.0	22.3%	0.0001
All—GT Pathological	70.3 ± 10.3	63.3 ± 8.6	10.0%	0.016
Experienced	66.6 ± 11.2	55.3 ± 8.7	17.0%	0.0065
Experienced—GT Normal	56.9 ± 7.2	46.2 ± 7.3	18.2%	0.0071
Experienced—GT Pathological	76.2 ± 13.9	64.3 ± 10.2	15.6%	0.021
Inexperienced	63.4 ± 7.5	54.4 ± 5.5	13.9%	0.017
Inexperienced—GT Normal	62.3 ± 8.4	46.3 ± 2.7	25.7%	0.0001
Inexperienced—GT Pathological	64.4 ± 6.7	62.3 ± 7.1	3.0%	0.29

## Data Availability

For proprietary reasons, the computer code for the anomaly detection tool cannot be made publicly available as it is proprietary to DeepC GmbH. The data that support the findings of this study are available from the corresponding author, (T.F.), upon reasonable request.

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
