# Peer review of "Faster and Better: How Anomaly Detection Can Accelerate and Improve Reporting of Head Computed Tomography"

_diagnostics, 2022, doi:10.3390/diagnostics12020452_

Round 1

Reviewer 1 Report

This is an interesting paper investigating the efficacy of an AI tool anomaly detection for identifying pathology in routinely acquired head CTs (for triage). In my opinion the research is rigorously conducted and the conclusions are solid.

I can’t appreciate the benefits brought by the AI tool in what it concerns the reduced investigation time as table 2 is missing from the manuscript.

Otherwise, the seems to me to be a good one, suitable for publication.

Author Response

Comments from reviewer 1

This is an interesting paper investigating the efficacy of an AI tool anomaly detection for identifying pathology in routinely acquired head CTs (for triage). In my opinion the research is rigorously conducted and the conclusions are solid.

Comment: I can’t appreciate the benefits brought by the AI tool in what it concerns the reduced investigation time as table 2 is missing from the manuscript.

Answer: Table 2 can be found on page 8/11, lines 271-278. In the initial version, the layout of the manuscript might have been corrupted as the table was wider than the document margins. We made appropriate corrections and now integrated both, figures and tables in adequate size, in an effort to enhance the readability of our manuscript.

Reviewer 2 Report

Authors present a novel AI-based anomaly detection tool for head computed tomography (CT) to provide patient-level triage and voxel-based highlighting of pathologies.

The paper is clear and well written. The Introduction including the Background, could be enriched to provide a synthetic overview of the state of the art of the decision support systems based on AI methods applied to medical imaging (or closer field to the head CT pathology detection if any).

Materials and Methods are clear, but I'd suggest to better describe how the "normal", "inconclusive" and "pathological" threshold are calibrated on an independent validation dataset.

Author Response

Comments from reviewer 2:

Authors present a novel AI-based anomaly detection tool for head computed tomography (CT) to provide patient-level triage and voxel-based highlighting of pathologies.

1st Comment: The paper is clear and well written. The Introduction including the Background, could be enriched to provide a synthetic overview of the state of the art of the decision support systems based on AI methods applied to medical imaging (or closer field to the head CT pathology detection if any).

Answer: Thank you for your valuable comment. We initially refrained from giving a too elaborate overview of existing CAD-systems in radiology in an effort to restrict the wordcount. However, we completely agree that providing a short overview of existing systems could clarify the background of our study. Therefore, we modified the introduction section accordingly:

Page 2, lines 50 - 57: “Multislice imaging in particular has recently witnessed the deployment of various decision-support systems, mainly aimed at flagging emergency findings, such as large-vessel occlusion in suspected stroke patients, or screening procedures such as CT-based lung cancer detection (6, 7). While excellent diagnostic performance was reported for these systems, they mainly consider single-class segmentations and thus remain of limited clinical utility in an unselected cohort. Multi-class systems, on the other hand, have been proposed for less data-intensive imaging modalities, such as chest X-ray and hip radiographs, that are generally readily interpretable by the radiologist (8, 9).”

2nd Comment: Materials and Methods are clear, but I'd suggest to better describe how the "normal", "inconclusive" and "pathological" threshold are calibrated on an independent validation dataset.

Answer: Thank you for this comment. Indeed, we agree that the quality of the manuscript could benefit from a more thorough explanation on how anomaly scores translate into patient-level labels. Therefore, we added explanations on how these thresholds were calibrated and hope to give the reader a better understanding on the threshold-based differentiations in labels.

Page 3, lines 105 - 118: “Outlier voxels were identified by comparing regions in the study set against the voxel-wise upper and lower bounds of the confidence interval from the training step. The ratio of outlier voxels to the entire brain volume was used to determine the patient level anomaly score, ranging from 0 to 1. Based on this anomaly score, patients were categorized into three classes ("normal", "inconclusive", "pathological") via thresholding. These thresholds were scanner-specific and have previously been calibrated on an independent validation set (not included in this study) of 61 CT scans (globally labelled as "normal" or "pathological") from the local CT scanner to minimize the false positive rate under the constraint of a false omission rate of 0, as explained in (10). If the anomaly score was higher than the pre-determined upper threshold (s > T-upper), the scan was labelled as “pathological” and anomalous voxels were added to the heatmap. Anomaly scores below the pre-determined lower threshold (s < T-lower), on the other hand, translated into “normal” labels for the scan, while anomaly scores between T-upper and T-lower led to an “inconclusive” label.”